# Phylogenetic and Comparative Genomics Study of *Cephalopina titillator* Based on Mitochondrial Genomes

**DOI:** 10.3390/insects16010006

**Published:** 2024-12-26

**Authors:** Huaibing Yao, Wanpeng Ma, Zhanqiang Su, Yuanyuan Yan, Yang Li, Weidong Cui, Jie Yang, Faqiang Zhan, Min Hou

**Affiliations:** 1Xinjiang Laboratory of Special Environmental Microbiology, Institute of Microbiology, Xinjiang Academy of Agricultural Sciences, Urumqi 830091, China; yaohuaibing@stu.xju.edu.cn (H.Y.); yuanyuanyan@webmail.hzau.edu.cn (Y.Y.); iamliyang91@163.com (Y.L.); cuwedo@163.com (W.C.); 2College of Veterinary Medicine, Xinjiang Agricultural University, Urumqi 830052, China; mwp086010@163.com (W.M.); szq00009@163.com (Z.S.); 3Key Laboratory of Biological Resources and Genetic Engineering, College of Life Science and Technology, Xinjiang University, Urumqi 830017, China; yangjie234@xju.edu.cn

**Keywords:** camel nasal botfly, *Cephalopina titillator*, mitogenomes, genetic analysis

## Abstract

*Cephalopina titillator* (Clark, 1816) (Diptera: Oestridae) is a common obligate parasite of camels, being distributed worldwide. In this study, we report the complete mitochondrial genome (mitogenome) of *C. titillator* and further investigate the phylogenetic relationships among other members of the Oestridae family. The results show that the mitogenome of *C. titillator* has a total size of 16,552 bp, exhibits a circular arrangement, and contains 13 PCGs, 22 tRNAs, two rRNAs, and two non-coding regions. A mitochondrial genome and phylogenetic analysis reveals the position of *C. titillator* in the Oestrinae subfamily of insects in the order Diptera, highlighting non-monophyly in this subfamily and identifying the Junggar camel isolate as a sister group to the previously reported Alxa camel isolate. The present study provides essential data for the phylogenetic positioning and molecular characterization of *C. titillator* and will serve as a foundational reference for further studies on camel bot flies and related Oestridae species.

## 1. Introduction

Bot flies (Calyptratae: Oestridae) are obligate parasites of mammals, with larvae that can cause myiasis in livestock, wildlife, and humans. The Bactrian camel (*Camelus bactrianus*), a vital livestock animal in arid desert regions, plays a crucial economic role in these environments. However, camels are frequently infected by various parasitic agents, impacting their health, as well as the quality and yield of camel milk and meat [1]. The larvae of *C. titillator* primarily parasitize the pharyngeal and nasal cavities of host camels, impairing their breathing and sometimes causing stridor, epithelial tissue damage, and even fatal secondary infections, such as meningitis. Adult *C. titillator* flies are also known to mechanically transmit pathogens. Several diagnostic methods, including serological and molecular testing of respiratory swabs, are available to detect these parasites in camels. However, diagnosing infestations in live camels remains challenging [2,3,4]. Field detection is further complicated as camels are often raised in remote desert areas where advanced diagnostic techniques may be impractical and distinguishing between similar neurological or respiratory symptoms caused by other pathogens can be difficult [5]. In recent years, research on *C. titillator* infestations in camels has grown, although no comprehensive studies map the current global distribution of these flies. The geographical distribution of *C. titillator* largely corresponds to their animal hosts, which are mostly concentrated in Central Asia and the Middle East, regions that are home to Bactrian and dromedary camels due to their arid and semi-arid environments, which also support various ectoparasites and endoparasites [6,7].

Currently, *C. titillator* larvae isolated from camels worldwide show consistent morphological characteristics. However, the genetic relationships and transmission dynamics between *C. titillator* infestations in Bactrian and dromedary camels worldwide remain unclear. Phylogenetic and evolutionary studies of *C. titillator* have long been hindered by the lack of complete mitochondrial genome (mitogenome) data, with most published sequences limited to short nucleotide fragments, primarily from the *cox1*, *cytb*, and *28S* rRNA genes [8,9,10,11]. To date, only one complete *C. titillator* mitogenome has been sequenced, derived solely from the Alxa Bactrian camel [12], while no mitogenomes have been reported from *C. titillator* in dromedary camels. In prior studies, our team identified *C. titillator* infestations in Bactrian from Xinjiang through a combination of analyses of microscopic morphology and the *cytb* and *cox* genes. These studies revealed the high prevalence of *C. titillator* in Bactrian camels, which is closely linked to factors such as age, season, pasture environment, and husbandry practices [9]. Given this context, it is essential to integrate multiple markers from mitogenomes to thoroughly investigate the taxonomic and phylogenetic relationships of *C. titillator* [1,12].

The mitogenome of insects represents an informative model for phylogenetics, molecular evolution, and comparative genomic research. Although substantial molecular sequence data and phylogenetic analyses exist for parasites, relatively few studies have focused on the molecular biology of myiasis-producing larvae, particularly within the family Oestridae [13]. Identifying *C. titillator* larvae can be challenging due to significant morphological similarities across subfamilies; however, molecular methods enable accurate identification across all larval stages [14]. This study reports the presence of *C. titillator* in Junggar Bactrian camels in Xinjiang, China, aiming to analyze potential regional differences in *C. titillator* infestations of the laryngopharyngeal areas of different camel breeds in two major camel-breeding provinces through mitogenome sequencing. Through this approach, we aim to obtain a comprehensive understanding of the complete mitogenome of *C. titillator*, clarify its potential phylogenetic status, and provide foundational data for exploring the phylogenetic relationships of *C. titillator* infestations in dromedaries in the Middle East. Overall, this study aims to enhance our understanding of *C. titillator* mitogenome diversity.

## 2. Materials and Methods

### 2.1. Larvae Sample Collection and Identification

Between 2019 and 2021, third-stage larval samples were collected from Junggar Bactrian camels slaughtered at the abattoir in Fuhai County, Altay Prefecture, China (87°31′37″ E, 47°6′42″ N). Fuhai County, located in the mid-temperate continental arid climate zone, is a major camel breeding base in Xinjiang. The infested camels were maintained by local herdsmen who traditionally graze on native pastures without feed supplements. The larvae were isolated from the nasopharynx of these camels. The specimens were morphologically identified and confirmed based on previous studies (Figure 1) [9,15,16]. The animal study was reviewed and approved by the Ethics Committee of Xinjiang Agricultural University (Ethics Number: 2023017).

### 2.2. DNA Extraction, Sequencing, and Mitogenome Assembly

Total genomic DNA of the collected larvae was extracted using a commercial Ezup column animal genomic DNA purification kit (Sangon Biotech, Shanghai, China) following the manufacturer’s instructions and stored at −20 °C. A genomic DNA library was constructed using the Illumina TruSeq DNA Sample Preparation Kit (Illumina, San Diego, CA, USA). The Illumina NovaSeq 6000 platform was employed for 150 bp paired-end sequencing performed by Shanghai Honsun Biological Technology Co., Ltd. (Shanghai, China). Low-quality bases and adapters from the merged reads were trimmed using Trimmomatic software v. 0.39 [17]. GetOrganelle v. 1.7.5 [18] was used to assemble the complete mitogenome data with the “animal_mt” default parameter. The sequencing coverage and depth are summarized in Appendix A.

### 2.3. Mitogenome Annotation and Characteristics Analysis

Gene annotations were refined by manual comparison with the mitogenomes of closely related species. Coding proteins and tRNA and rRNA genes were predicted using the MITOS2 web server [19]. The secondary structure prediction of the tRNA gene was also performed by MITOS2 and visualized using VARNA v. 3 [20]. The complete circular map of the mitogenome was drawn using the CGView visualization software v. 2.0.3. Relative synonymous codon usage (RSCU) was computed using cusp (EMBOSS v. 6.6.0.0). The codon usage preferences are listed in Appendix A. Strand asymmetry was evaluated by adenine–thymine (AT) and guanine–cytosine (GC) skews using the formulae AT skew = [A% − T%]/[A% + T%] and GC skew = [G% − C%]/[G% + C%], respectively [21,22]. The mitogenome was deposited in GenBank (accession number: PQ557251).

### 2.4. Mitogenome Collinearity and Gene Rearrangement Analysis

To better understand mitogenome evolution, a mitogenome collinearity analysis was performed among selected representative genomes within the family Oestridae to detect the homology and conservation characteristics of the mitogenome. Collinearity analysis was performed using Mauve software v. 2.4.0 [23]. Subsequently, packages such as ggplot2 and dplyr in R 4.4.1 were used to create custom scripts for drawing gene rearrangement maps of the mitogenome, displaying the starting and ending positions of genes in different species, and comparing gene rearrangements between species.

### 2.5. Phylogenetic Analysis and Tree Generation

The phylogenetic relationships among Oestridae were investigated using one newly sequenced sample from this study and nine sequences of known species (Table 1). Four species from *Calliphora* (Calliphoridae) and *Sarcophaga* (Sarcophagidae) were selected as outgroups according to previous phylogenetic analyses [12]. Our dataset was based on the nucleotide sequences of 13 mitochondrial protein-coding genes (PCGs) (*nad2*, *cox1*, *cox2*, *atp8*, *atp6*, *cox3*, *nad3*, *nad5*, *nad4*, *nad4l*, *nad6*, *cob*, and *nad1*), rRNA, and tRNA genes. All phylogenetic analysis steps were conducted using PhyloSuite v. 1.2.3 and its plug-in programs [24]. Nucleotide sequences for each gene were aligned using MAFFT v. 7.505 software [25]. The 13 protein-coding regions were aligned in codon mode using the ’auto’ alignment parameters, and the remaining 24 RNA-coding genes were aligned using the default parameters. Gaps of multiple sequence alignments were eliminated using trimAl v.1.2 [26] using the “automated1” option. Subsequently, the trimmed sequences were aligned and concatenated. Several partitioning schemes were tested, including the division of protein-coding genes into the 1st, 2nd, and 3rd codon positions. According to the AICc criterion, we used ModelFinder v. 2.2.0 [27] to select the optimal substitution models for each partition of IQTree and MrBayes. The model results are shown in Appendix A. Maximal likelihood phylogenetic analysis was performed using IQ-TREE v. 2.2.0 [28] with ultrafast bootstrapping (10,000 replicates). MrBayes v. 3.2.7 [29] was used to reconstruct a Bayesian Inference (BI) tree. The chain length was set to 2,000,000 with a log at every 1000 units, and the Markov chain Monte Carlo (MCMC) algorithm was run twice. Parameter convergence and effective sample size (ESS > 200) were assessed using Tracer v. 1.7.2 [30]. The first 50% of trees were discarded as burn-in, and the remaining trees were used to generate the consensus tree. Finally, the phylogenetic trees were visualized and enhanced in FigTree v. 1.4.4 [31]. Mitochondrial genome sequence alignment was performed using Muscle v. 3.8.31 to validate the phylogenetic analysis [32], and pairwise identity among sequences was validated using a sequence demarcation tool (SDT v. 1.2) [33].

## 3. Results

### 3.1. Mitogenome Organization and Nucleotide Composition

The complete mitogenome of *C. titillator* was 16,552 bp in length, with an average gene length of 861 bp (Figure 2). The genome comprised 37 typical mitogenome genes, including 22 tRNA genes, 13 PCGs, two rRNA genes, and two non-coding regions for the origin of the light strand (OL) and control region (CR) (Table 2). Among these genes, 13 genes with four PCGs (*nad4*, *nad4l*, *nad5*, and *nad6*), two tRNA genes (*rrnL* and *rrnS*), and seven tRNA genes (*trnQ-ttg*, *trnC-gca*, *trnY-gta*, *trnH-gtg*, *trnT-tgt*, *trnL1-tag*, and *trnV-tac*) were encoded on the minority strand (R), whereas 24 genes, including nine protein-coding genes and 15 tRNA genes, were located on the majority strand (H; Table 2). Moreover, the analysis showed seven overlapping regions in the mitogenome of *C. titillator*, with a total length of 17 bp and varying in size from 1 to 6 bp. The longest overlapping regions were located between *atp6* and *atp8* and between *nad4* and *nad4l* genes. The stop codons were represented by TAA, TAG, or a single T, where four PCGs (*cox2*, *nad3*, *nad4*, and *nad5*) were terminated by the stop codon of a single T, four PCGs (*cox1*, *nad1*, *nad2*, and *cob*) were terminated by TAG, and the remaining five PCGs (*atp8*, *atp6*, *cox3*, *nad4l*, and *nad6*) were terminated by TAA. Furthermore, we observed a gap between *tRNAI* and *tRNAQ*, which is consistent with previous sequencing results [12].

Further analysis showed that the complete mitogenome of *C. titillator* contained 13 PCGs with a total length of 11,189 bp, accounting for 67.59% of the mitogenome (Table 3). The PCGs consisted of seven NADH dehydrogenases, three cytochrome c oxidases, two ATP synthases, and one cytochrome b. The nucleotide composition of the complete mitogenome of *C. titillator* was 41.12%, 32.19%, 17.82%, and 8.87% for bases A, T, C, and G, respectively. The A + T content (73.31%) was greater than the G + C content (26.69%), indicating an evident AT bias. This A + T content was lower than the reported 76% in Oestrinae in the literature [12]. Additionally, for the complete mitogenome, the AT skew had a positive value (AT skew = 0.12), while the GC skew had a negative value (GC skew = −0.34), indicating the higher abundance of A than T and of C than G.

### 3.2. Protein-Coding Genes and Codon Usage

Analysis of relative synonymous codon usage (RSCU) in the *C. titillator* mitogenome (Figure 3) showed that 28 codons had RSCU values greater than or equal to 1.0, and 33 codons had RSCU values lower than 1.0. A preferred codon was defined as one with an RSCU value greater than 1.0. Among these, 14 had RSCU values greater than or equal to 1.6. The most commonly used codons were TTA (Leu2), TCT (Ser), and GGA (Gly), while Ala, Arg, Gly, Leu, Pro, Ser, Thr, and Val were the most frequently encoded amino acids. Additionally, most of the 24 preferred codons in this study ended with the A/T base, accounting for 93.33% of the codons, indicating that frequently used codons tended to end with this base. Appendix A shows the predicted secondary tRNA structure. Of these 22 tRNA genes, 21 tRNAs could be folded into a typical cloverleaf secondary structure. The trnaS1 (Ser) lacked the dihydrouridine arm (Figure 4). This phenomenon was consistent with previously reported mitogenomes of the subfamily Oestridae.

### 3.3. Mitogenome Collinearity and Gene Rearrangement

To analyze the structures and compositions of *C. titillator* larval mitogenomes from Xinjiang Junggar Bactrian camels, a mitogenome collinearity analysis was performed with the mitogenomes of other subfamilies, showing that the *C. titillator* of Junggar Bactrian camels, *Cephalopina trompe* of reindeer, *Rhipicephalus usbekistanicus* of equids, and *C. titillator* of Alxa Bactrian camels exhibited the same genome organization (Figure 5). These results illustrate that the *C. titillator* mitogenome had good collinearity with the reference genome, indicating the high quality and accuracy of the map. In addition, gene order data were compared to examine the gene rearrangements. Comparative mitogenome analysis revealed that species from different subfamilies had the same gene order. Mitogenomic rearrangements of PCGs were not observed among the bot flies, indicating no gene recombination (Figure 6).

### 3.4. Phylogenetic Analysis

To clarify the phylogenetic relationships among the numerous significant subfamilies of Oestridae, we constructed maximum likelihood (ML) and BI phylogenetic trees. The bootstrap and branch values were high in two phylogenetic trees, exhibiting high confidence levels. The phylogenetic reconstructions of MT genes revealed similar topologies (Figure 7a,b), with the majority of nodes having 100% bootstrap values, exhibiting 1.00 Bayesian posterior probabilities, and being highly facilitative to the monophyly of the family Oestridae. The evolutionary trees illustrated the positions of different species within Oestridae after adding the new sequences of *C. titillator* from Xinjiang, China. The phylogenetic trees of the two halves of the alignment differed only in terms of the placement of *Dermatohia hominis*. The phylogenetic analysis of *C. titillator* mitogenome classified it as Oestrinae; however, the subfamily Oestrinae did not exhibit monophyly. The two phylogenetic trees showed that *C. titillator* was clustered into the same branch, verifying the accuracy of the results. Moreover, the pairwise sequence identity of *C. titillator* sequences from Xinjiang with MN833258.1 *C. titillator* reported from Inner Mongolia was > 98% (Figure 8). The gene order and overall structure of the *C. titillator* mitochondrial genome were consistent with the reference genome sequence (MN833258.1), confirming that they belonged to the same species. However, *C. titillator* isolated specimens from the Junggar and Alxa Bactrian camels were distributed in different groups of the same branch. The aforementioned findings indicated that the Oestrinae subfamily did not exhibit monophyly, especially when considering the newly added *C. titillator*, which showed sister lineage relationships with *C. titillator* (MN833258.1). This observation underscored the pronounced genetic disparities between these distinct species.

## 4. Discussion

Recently, the widespread use of molecular techniques, particularly DNA sequencing, has significantly advanced insect taxonomy [43]. This is particularly important for the endoparasitic larval stages of myiasis-causing dipteran insects that infest the internal tissues and organs of animals. The coevolution of host and parasites is a major component of the evolution of life [44]. The life histories of the host animals seem to be the driving force for evolution within these parasitic flies [45]. *C. titillator* is an obligate parasite that exclusively infests camels belonging to the family Camelidae, with its larvae causing myiasis. Despite its high prevalence, there is limited research on the genetic variability, molecular evolution, and phylogenetic relationships of *C. titillator.* Previous studies have suggested that the DNA sequences of the *cytb* and *cox1* genes in *C. titillator* infestations in Bactrian and dromedary camels from different regions are highly similar [1,9,10]. However, these studies have been based on partial mitochondrial gene sequences, restricting the scope of the analysis.

Mitochondrial genomes of insects have been extensively studied based on their phylogenetic and evolutionary biology studies. Notably, there has been no report on the mitochondrial genome of *C. titillator* from Junggar Bactrian camels. Therefore, the present study provides the first complete sequencing of the mitogenome of *C. titillator* from Xinjiang, China, and compares it to the mitogenomes of other Oestrinae species. Our results show that the mitogenome of *C. titillator* has a closed circular structure with a total size of 16,552 bp, which is consistent with previous findings [12]. The mitogenome is composed of 37 genes (13 PCGs, 22 tRNAs, and two rRNAs), an OL, and a CR. The genome organization and composition are in accordance with those of most bot fly species reported previously. All genes are arranged in the same order and direction as those in the ancestral insect mitogenome [46]. AT and GC skews are measures of compositional asymmetry. In the present study, the mitogenome A + T content is higher than the mitogenome G + C content. Moreover, A is more abundant than T, with an AT skew of 0.12, and C is more abundant than G, with a GC skew of −0.34. Several studies have demonstrated that after birth, almost all animals, including insects, are characterized by an AT bias, indicating that AT-rich regions may represent the origin of replication and may be more easily altered during evolution [44]. Additionally, due to the limited resources for nucleotide production, the synthesis of G + C consumes more energy and nitrogen than that of A + T, making A and T the preferred nucleotides.

The start codons for the 13 PCGs are ATG, ATA, ATT, GTG, and TCG. Most PCGs are represented by the complete stop codons of TAA or TAG. Specifically, four PCGs (*cox2*, *nad3*, *nad4*, and *nad5*) have incomplete stop codons consisting of a single T. Incomplete stop codons differ among different taxa [47]. Incomplete stop codons are commonly observed in dipteran species. RSCU represents the characteristics of codon usage bias in the mitogenome. The results of the RSCU analysis of PCGs in the mitogenome of *C. titillator* indicate that the most frequently used codons are TTA, TCT, and GGA, encoding the amino acids Leu2, Ser, and Gly, respectively. As *C. titillator* specifically parasitizes camels, codon usage analysis of PCGs in its mitogenome is important for elucidating its origin and evolution. Genome collinearity analysis reveals that the collinearity between the genomes of *C. titillator* from Xinjiang and that from Inner Mongolia is much higher than that of other species. Gene rearrangement is considered a valuable phylogenetic signal for studying mitochondrial evolution [48]. Gene rearrangement analysis corresponds with mitogenome collinearity analysis. The intergenic and overlapping regions of *C. titillator* are similar to most mitogenomes of Oestridae, with no gene rearrangements. Additionally, no closely related genus or species within these subfamilies has been identified with similar gene rearrangements to date.

It is widely accepted that animal mitogenome has a high evolutionary rate and lacks genetic recombination [49,50]. The mitochondrial genome is haploid and maternally inherited and is characterized by differences in the rate of nucleotide substitutions in different genes. The genes in the mitochondrial genome have a certain degree of resemblance among different species, i.e., their sequences and structures are similar in different species. However, the limitations include differences in evolutionary rates, genome rearrangements, and heterogeneity, as well as limitations in analysis methods and models, technical constraints, and cost issues. In the present study, rooted phylogenetic trees are used to elucidate the phylogenetic relationships between *C. titillator* larvae amplicons and those of other members of the family Oestridae obtained from GenBank. To better determine the phylogenetic position of *C. titillator* from Xinjiang and further clarify the evolutionary relationships, the phylogenetic analyses are based on 18 mitochondrial genomes, using *Calliphora* (Calliphoridae) and *Sarcophaga* (Sacophagidae) as outgroups. The phylogenetic topologies of the ML and BI method are similar (Figure 7), and most nodes are supported by high values. The mitochondrial genome phylogeny shows that *C. titillator* from Xinjiang is of a sister lineage of *C. titillator* larvae from Inner Mongolia. Previous research has illustrated that Egyptian *C. titillator* is a sister group to the subfamily Gasterophilinae based on *cox1* and *28S* rRNA genes [1], which is consistent with our findings. The present study suggests a closer relationship between Oestrinae and Gasterophilinae; however, more whole-genome sequences are necessary to elucidate the paraphyly and phylogenetic diversity within the family. Phylogenetic reconstructions based on 13 mitochondrial PCGs and RNA genes provide a relatively robust phylogenetic framework for *C. titillator*, especially for its various hosts, improving our understanding of the phylogenetic relationships of Oestridae.

Parasites exhibit varying degrees of host specificity, reflecting their unique coevolutionary histories with host species [51]. Since *C. titillator* infestations in migratory camels are subjected to multiple environmental and biotic changes along their migration routes, migration may enhance the cophylogenetic congruence between camels and *C. titillator*. However, the role of mitogenomes in determining the host specificity of *C. titillator* remains unclear. Currently, *C. titillator* is specialized to a single host, the camel, and does not infest other species. As a result, the phylogenetic relationships of *C. titillator* across different camel species remain unknown. Future studies should aim to expand the sample size and include a broader range of collection locations to gain a deeper and more comprehensive understanding of the phylogenetic relationships of *C. titillator* in both Bactrian and dromedary camels. In addition, it is important to note that in Asia and the Middle East, *C. titillator* has a significant impact on camel health and productivity. Therefore, effective preventive measures should be implemented in camel breeding and management to control the occurrence and spread of this parasitic disease.

## 5. Conclusions

The present study reports the complete mitogenome of *C. titillator* larvae obtained from Xinjiang Junggar Bactrian camels. The total size of the mitogenome is 16,552 bp, comprising 13 PCGs, 22 tRNAs, two rRNAs, and two non-coding regions for OL and CR. Its length is shorter than that of the mitogenome of *C. titillator* isolated from the Alxa Bactrian camel. The mitogenome also shows clear AC bias. The mitogenome of *C. titillator* is conserved in structure, and the genes retain the same order and direction as the ancestral insect mitogenome. Our molecular analyses of *C. titillator* from Xinjiang indicate that this species is a sister lineage of *C. titillator* from Inner Mongolia. Furthermore, we have found that the phylogenetic relationships among *C. titillator* from different regions are closely related to host camel migration. Future studies should broaden the range of sampling areas and place greater emphasis on different camel breeds and wild camel populations. Altogether, our mitogenome sequencing analysis provides a comprehensive overview of the taxonomic status of *C. titillator* and is of great value for explaining the evolutionary relationships within the family Oestridae.

## Figures and Tables

**Figure 1 insects-16-00006-f001:**
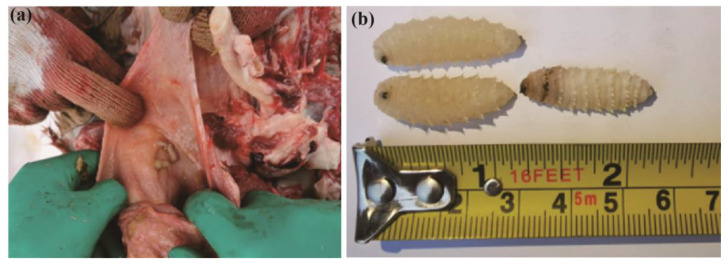
Images of the larval specimens of *Cephalopina titillator*: (**a**) post-mortem view of heavy infestation with *C. titillator* larvae in the pharyngeal region and (**b**) measurements of the larval body.

**Figure 2 insects-16-00006-f002:**
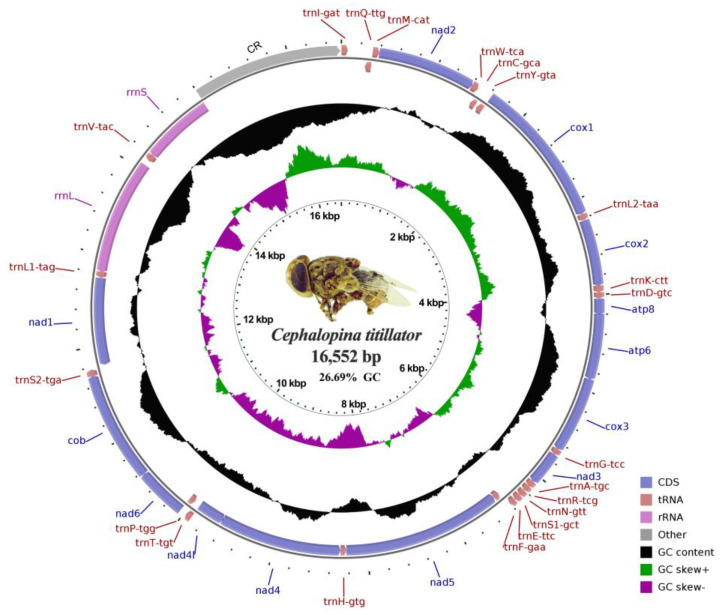
A circular map of the mitogenome of *Cephalopina titillator*. The arrows indicate the direction of gene transcription. The gene lengths are proportional to their nucleotide lengths.

**Figure 3 insects-16-00006-f003:**
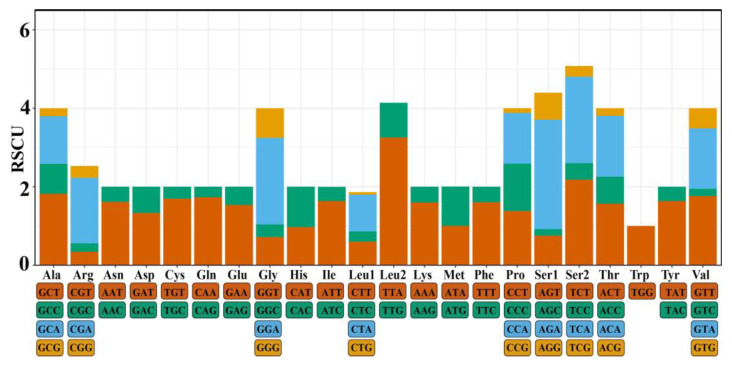
Relative synonymous codon usage (RSCU) of the mitogenome of *Cephalopina titillator*.

**Figure 4 insects-16-00006-f004:**
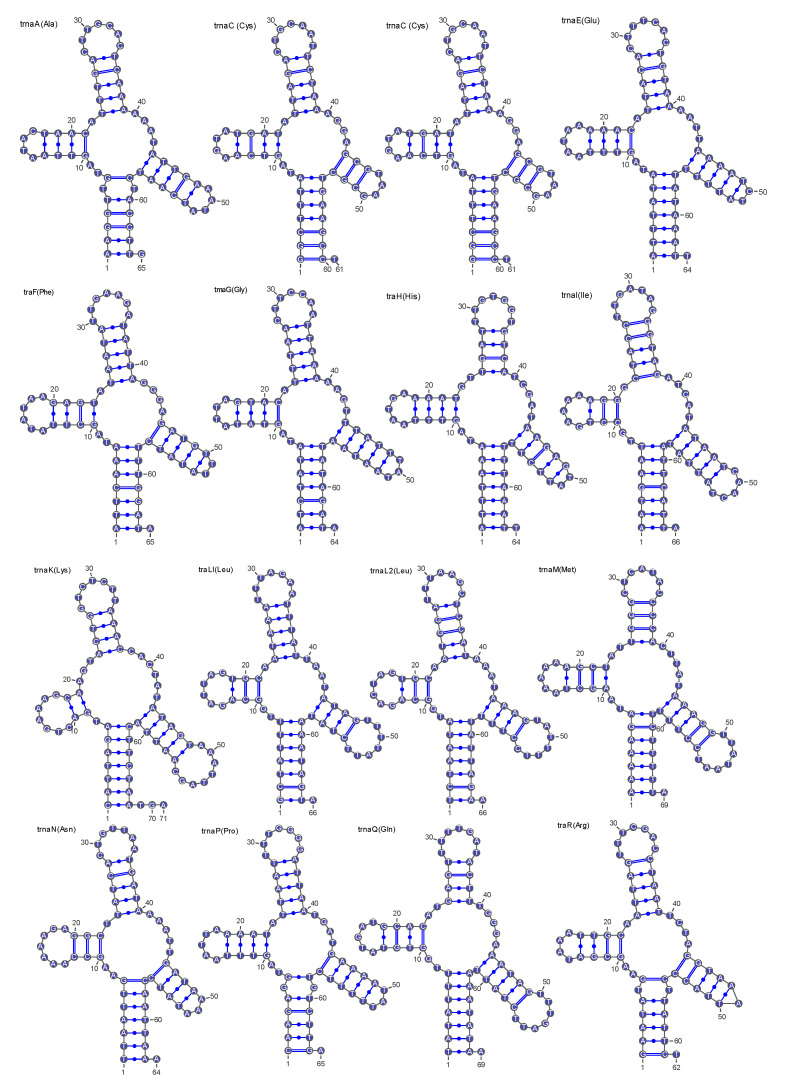
The inferred secondary structures of tRNAs in the complete mitochondrial genome of *Cephalopina titillator*. The 22 tRNA genes were marked with standard abbreviations in the top left, and the corresponding amino acids for translocation were listed in parenthesis.

**Figure 5 insects-16-00006-f005:**
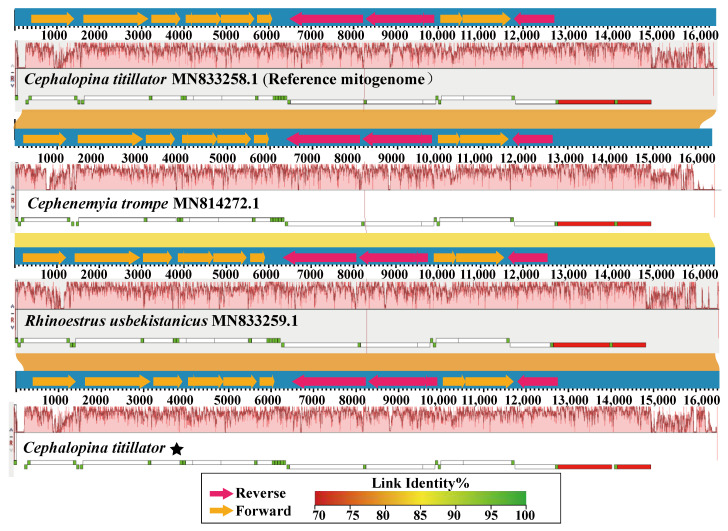
Progressive Mauve analyses showing the genome size variations and global rearrangement structures of four Oestridae mitogenomes. Each genome is presented horizontally, and homologous segments are shown as colored blocks connected across the genomes. The stars represent the species sequenced in this study.

**Figure 6 insects-16-00006-f006:**
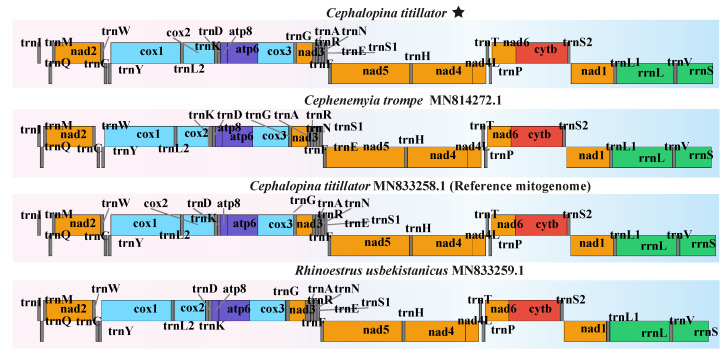
The mitogenome rearrangement patterns of *C. titillator* compared with the reference mitogenomes. The stars represent the species sequenced in this study.

**Figure 7 insects-16-00006-f007:**
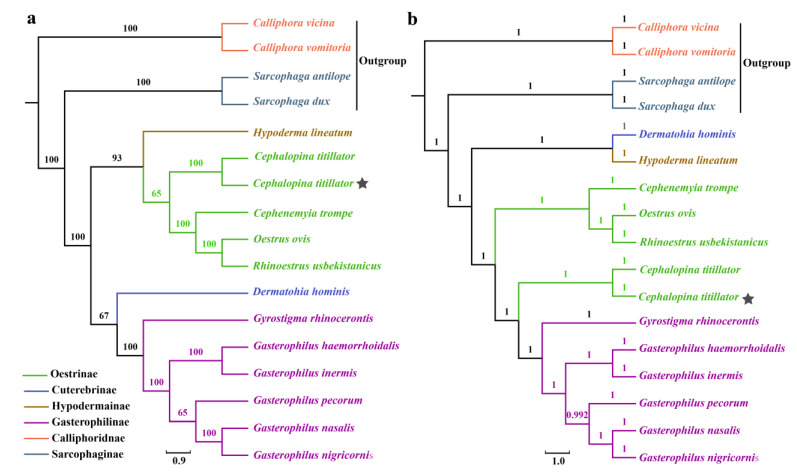
The Oestridae phylogenetic relationships inferred from the mitochondrial genomes. (**a**) A phylogenetic tree constructed using the nucleotide sequences of 13 PCG, two rRNA, and 22 tRNA genes using the maximum likelihood (ML) method. (**b**) A Bayesian phylogenetic tree of *C. titillator* from Xinjiang. Different colors in the phylogenetic tree represent different branches. Numerical values on the nodes represent the bootstrap confidence, and the scales denote evolutionary distance. Stars represent the species sequenced in this study.

**Figure 8 insects-16-00006-f008:**
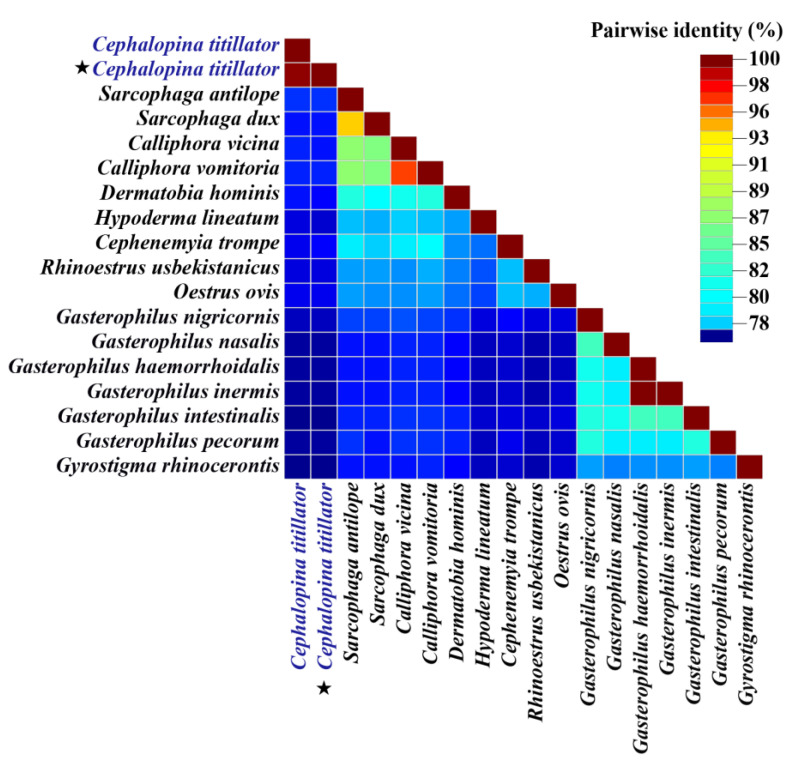
A color-coded pairwise identity matrix generated from 18 sample mitochondrial genome sequences. Each colored cell represents a percentage identity score between two sequences (left horizontal and bottom vertical). The colored key in the matrix indicates the percentage pairwise identity. The stars represent the species sequenced in this study.

**Table 1 insects-16-00006-t001:** Information on the mitogenome used for phylogenetic analysis in this study.

Family	Subfamily	Species	Host Animals	GenBank Accession No.	References
Oestridae	Oestrinae (Nasopharyngeal Myiasis)	*Cephalopina titillator*	Camel	PQ557251.1	Present study
		*Cephalopina titillator*	Camel	MN833258.1	[12]
		*Cephenemyia trompe*	Reindeer	MN814272.1	[12]
		*Oestrus ovis*	Sheep	NC_059851.1	[34]
		*Rhinoestrus usbekistanicus*	Equidae	MN833259.1/NC_045882.1	[12]
	Cuterebrinae (Cutaneous Myiasis)	*Dermatobia hominis*	Human	NC_006378.1	[35]
	Gasterophilinae (Gastrointestinal Myiasis)	*Gasterophilus intestinalis*	Equidae	MG920504.1	[36]
		*Gyrostigma rhinocerontis*	Rhinocerotidae	MK045312.1	[36]
		*Gasterophilus haemorrhoidalis*	Equidae	MG920502.1	[36]
		*Gasterophilus inermis*	Equidae	MG920503.1	[36]
		*Gasterophilus nasalis*	Equidae	MG920505.1	[36]
		*Gasterophilus nigricornis*	Equidae	MG920506.1	[36]
		*Gasterophilus pecorum*	Equidae	KU578262.1	[37]
	Hypodermainae (Cutaneous Myiasis)	*Hypoderma lineatum*	Cattle	GU584123.1	[38]
Calliphoridae	Calliphoridnae	*Calliphora vicina*	Not given	JX913760.1	[39]
		*Calliphora vomitoria*	Not given	KT444440.1	[40]
Sarcophagidae	Sarcophaginae	*Sarcophaga antilope*	Not given	MH540748.1	[41]
		*Sarcophaga dux*	Not given	MH937748.1	[42]

**Table 2 insects-16-00006-t002:** The positions and features of the genes in the *Cephalopina titillator* mitogenome.

Gene	Strand	Genes Position	Size (bp)	Intergenic Spaces (bp) Number	Start Codon	Stop Codon
*trnI-gat*	J	1–66	66	0	---	---
*trnQ-ttg*	N	253–321	69	186	---	---
*trnM-cat*	J	321–389	69	−1	---	---
*nad2*	J	390–1400	1011	0	ATT	TAG
*trnW-tca*	J	1399–1467	69	−2	---	---
*trnC-gca*	N	1474–1534	61	6	---	---
*trnY-gta*	N	1558–1626	69	23	---	---
*cox1*	J	<1625–3158	1534	−2	TCG	TAG
*trnL2-taa*	J	3159–3224	66	0	----	---
*cox2*	J	3227–3911	685	2	ATG	T--
*trnK-ctt*	J	3909–3979	71	−3	---	---
*trnD-gtc*	J	3979–4042	64	−1	----	---
*atp8*	J	4043–4204	162	0	ATT	TAA
*atp6*	J	4198–4875	678	−7	ATG	TAA
*cox3*	J	4875–5663	789	−1	ATG	TAA
*trnG-tcc*	J	5664–5727	64	0	---	---
*nad3*	J	5728–6079	352	0	ATT	T--
*trnA-tgc*	J	6080–6144	65	0	---	---
*trnR-tcg*	J	6144–6205	62	−1	---	---
*trnN-gtt*	J	6209–6272	64	3	---	---
*trnS1-gct*	J	6273–6338	66	0	---	---
*trnE-ttc*	J	6342–6405	64	3	---	---
*trnF-gaa*	J	6426–6490	65	20	---	---
*nad5*	N	6491–8222	1732	0	GTG	T--
*trnH-gtg*	N	8223–8286	64	0	---	---
*nad4*	N	8287–9622	1336	0	ATG	T--
*nad4l*	N	9616–9912	297	−7	ATG	TAA
*trnT-tgt*	N	9915–9979	65	2	---	---
*trnP-tgg*	J	9981–10,045	65	1	---	---
*nad6*	N	10,048–10,575	528	2	ATA	TAA
*cob*	J	10,575–11,711	1137	−1	ATG	TAG
*trnS2-tga*	J	11,710–11,777	68	−2	---	---
*nad1*	J	11,794–12,741	948	16	ATG	TAG
*trnL1-tag*	N	12,744–12,809	66	2	---	---
*rrnL*	N	12,808–14,072	1265	−2	---	----
*trnV-tac*	N	14,132–14,203	72	59	---	---
*rrnS*	N	14,202–14,994	793	−2	---	---

‘J’ and ‘N’ represent the majority and minority strands, respectively, and T indicates an incomplete stop codon.

**Table 3 insects-16-00006-t003:** Nucleotide composition of the *Cephalopina titillator* mitogenome, showing the nucleotide compositions of the whole genome, protein-coding genes (PCGs), rRNAs, tRNAs, two non-coding regions of the light strand (OL), and the control region (CR), with thirteen PCGs listed in rows.

Feature	Proportion of Nucleotides							
	Size (bp)	T%	C%	A%	G%	A + T (%)	AT Skew	GC Skew
Whole genome	16,552	32.19	17.82	41.12	8.87	73.31	0.12	−0.34
PCGS	11,189	40.33	15.63	29.58	14.46	69.91	−0.15	−0.04
First position	3729	41.88	15.36	26.6	16.14	68.48	−0.22	0.02
Second position	3729	40.22	17.35	29.95	12.47	70.17	−0.15	−0.16
Third position	3729	38.88	14.18	32.17	14.75	71.05	−0.09	0.02
tRNA	1454	36.8	10.94	37.83	14.44	74.62	0.01	0.14
rRNA	2058	42.81	6.56	36.1	14.53	78.91	−0.09	0.38
*16S rRNA*	1265	43.95	6.09	36.36	43.95	80.31	0.09	0.38
*12S rRNA*	793	40.98	7.3	35.69	40.98	76.67	−0.07	0.37
CR	1558	48.01	38.38	8.79	4.82	86.39	0.11	−0.29
OL	113	45.61	47.37	1.75	5.26	92.98	−0.02	−0.5
*atp6*	678	34.81	33.19	21.68	10.32	67.99	0.02	0.35
*atp8*	162	38.27	38.27	19.75	3.70	76.54	0.00	0.68
*cob*	1137	33.33	33.42	22.43	10.82	66.75	0.00	0.35
*cox1*	1534	31.62	33.12	20.73	14.54	64.73	−0.02	0.18
*cox2*	685	34.16	32.41	20.88	12.55	66.57	0.03	0.25
*cox3*	789	32.19	32.57	22.05	13.18	64.76	−0.01	0.25
*nad1*	948	20.68	50.42	8.33	20.57	71.10	−0.42	−0.42
*nad2*	1011	34.42	39.56	17.31	8.70	73.99	−0.07	0.33
*nad3*	352	32.67	36.65	21.59	9.09	69.32	−0.06	0.41
*nad4*	1336	22.90	49.85	7.78	19.46	72.75	−0.37	−0.43
*nad4l*	297	24.92	52.53	4.71	17.85	77.44	−0.36	−0.58
*nad5*	1732	23.96	48.21	7.91	19.92	72.17	−0.34	−0.43
*nad6*	528	39.02	36.74	17.99	6.25	75.76	0.03	0.48

## Data Availability

The original sequencing sequence data have been deposited in the NCBI Sequence Read Archive database, access number PRJNA1197122.

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
