# Peer review of "Phylogenetic and Comparative Genomics Study of Cephalopina titillator Based on Mitochondrial Genomes"

_insects, 2024, doi:10.3390/insects16010006_

Round 1

Reviewer 1 Report

Comments and Suggestions for Authors

In the paper by Yao et al. Phylogenetic and comparative genomic studies of Cephalopina tutillator based on mitochondrial gemones authors presents sequence of the mitochondrial genome of C. titillator. Moreover  they present also phylogenetic analysis.

Authors presents obtained results as it would be first decription of the mt genome for Cephalopina tutillator though mt genome for that species was described in 2020 by Li et al. (https://doi.org/10.1016/j.ijbiomac.2020.01.249.).

This is in fact first descrition for Xinjiang C.tutillator, that is different population that in paper from 2020, so in my opinnion it should focus on differences between those two genomes and I lack such comparison.

Detailed description of the mt genome especially if it was previously described does not bring any new knowledge (what new is is nucleotide composition of mitogenome?, What new infomation bring the inferred secondary structures of the tRNAs?)

I suggest focusing more on the differences between the mt genomes from these two populations if there are any. Further, analysis of the degree of heterogeneity should be performed to determine these differences.

Moreover, ML phylogenetic analyses must be corrected. There is no information on the branch support method that was used, and no information about parameters of gamma distribution and invariable sites since  the GTP+I+G model was used.

In the Phylogenetic analysis description, there is no information on whether the tree is rooted, what outgroup was used to root the topologies. I suggest preparing the analysis as in Li et al. (2020) and using the same outgroups, which is why there would be a proper comparison of the results.

A phylogenetic tree lacks a bar indicating the number of changes per site,   and this is a standard practice  and should be presented in a phylogenetic tree.

More, Bayesian interference analyses should be performed in addition to ML. It is standard to use both of these methods if possible.

Author Response

Response to Reviewer Comments

Dear reviewer:

Thanks for your professional advices. Following your suggestions, we have re-edited our manuscript throughout carefully and addressed all the comments in the notes below.

Please note that our responses to the reviewer’s comments are shown in red font, and line numbers mentioned here are shown in the manuscript.

Point 1:In the paper by Yao et al. Phylogenetic and comparative genomic studies of Cephalopina tutillator based on mitochondrial gemones authors presents sequence of the mitochondrial genome of C. titillator. Moreover, they present also phylogenetic analysis.

Authors presents obtained results as it would be first decription of the mt genome for Cephalopina tutillator though mt genome for that species was described in 2020 by Li et al. (https://doi.org/10.1016/j.ijbiomac.2020.01.249.).

This is in fact first descrition for Xinjiang C.titillator, that is different population that in paper from 2020, so in my opinnion it should focus on differences between those two genomes and I lack such comparison.

Reponse 1: Thanks for your comments. As you mentioned, as shown in Figure 5,Figure 6, and Figure 8 we conducted a comparative analysis of their mitochondrial genomes. Then there are subtle differences between them, but it is difficult to explain their reasons. Therefore, in future research, we expect to collect C.titillator samples from dromedary camels for comparative analysis.

Point 2:Detailed description of the mt genome especially if it was previously described does not bring any new knowledge (what new is is nucleotide composition of mitogenome?, What new infomation bring the inferred secondary structures of the tRNAs?)

Reponse 2: We deeply appreciate the guidance and feedback you've provided. At present, there is no mitochondrial genome sequencing analysis of C.titillator in Xinjiang Bactrian camels. This study reports for the first time the results of mitochondrial genome sequencing of Xinjiang Bactrian camels, which is itself an exploration of new knowledge. Although the nucleotide composition of the mt genome (A, T, C, G) is basic and universal in biology, this study accurately determined the nucleotide composition of the mt genome of Xinjiang Bactrian camel A and accumulated detailed data (A: 41.42%, T: 32.19%, C: 17.82%, G: 8.87%). These data not only provide basic data for the genomic research of this species, but also provide valuable references for comparing the nucleotide composition differences of mt genomes between different species. Although this information is not novel in biological principles, it is of great significance for understanding the genetic characteristics and evolutionary history of the species.

Previous studies did not predict and analyze the secondary structure of tRNAs in C.titillator.

Table Component_stat of C.titillator in Xinjiang Bactrian camels

Region

Length(bp)

T%

C%

A%

G%

Genome

16552

32.19

17.82

41.12

8.87

Protein_coding_genes

11189

40.33

15.63

29.58

14.46

First position

3729

41.88

15.36

26.6

16.14

Second position

3729

40.22

17.35

29.95

12.47

Third position

3729

38.88

14.18

32.17

14.75

tRNA

1454

36.8

10.94

37.83

14.44

rRNA

2058

42.81

6.56

36.1

14.53

We have added sentences for comparative analysis with previous literature.

“This is lower than the reported A+T content of 76% in Oestrinae in the literature.”

Point 3: I suggest focusing more on the differences between the mt genomes from these two populations if there are any. Further, analysis of the degree of heterogeneity should be performed to determine these differences.

Reponse 3:We appreciate your thoughtful comments and suggestions. As the results show, the mitochondrial genome differences between the two are relatively small. In future research, we will collect C.titillator from dromedary camels and combine these mitochondrial genome data for degree of heterogeneity analysis.

Point 4: Moreover, ML phylogenetic analyses must be corrected. There is no information on the branch support method that was used, and no information about parameters of gamma distribution and invariable sites since  the GTP+I+G model was used.

Reponse 4: Thanks for your suggestions. We have re-edited this paragraph. We have developed a partition selection model for the mitochondrial genome, which divides the protein coding region into independent partitions and selects the optimal model for phylogenetic analysis. The detailed model results can be found in the TableS2.

IQTree:

Subset partitions      Best model

P1: (rrnL_mafft_trimAl)  TIM3+F+R3

P2: (rrnS_mafft_trimAl)  TPM2u+F+R2

P3: (trnA_mafft_trimAl_trnM_mafft_trimAl_trnN_mafft_trimAl_trnR_mafft_trimAl_trnS1_mafft_trimAl_trnS2_mafft_trimAl)         TIM2+F+I+G4

P4: (trnC_mafft_trimAl)  TPM2u+F+G4

P5: (trnD_mafft_trimAl_trnG_mafft_trimAl_trnT_mafft_trimAl)    TIM2+F+I+I+R2

P6: (trnE_mafft_trimAl)  HKY+F+R2

P7: (trnF_mafft_trimAl_trnP_mafft_trimAl)    TPM3u+F+G4

P8: (trnH_mafft_trimAl)  TN+F+G4

P9:(trnI_mafft_trimAl_trnL2_mafft_trimAl_trnQ_mafft_trimAl_trnW_mafft_trimAl) TPM2u+F+R2

P10: (trnK_mafft_trimAl) HKY+F+I+G4

P11: (trnL1_mafft_trimAl_trnV_mafft_trimAl)        HKY+F

P12: (trnY_mafft_trimAl) TIM3+F+G4

P13: (atp6_mafft_trimAl_nad3_mafft_trimAl) GTR+F+R3

P14: (atp8_mafft_trimAl) HKY+F+I+G4

P15: (cox1_mafft_trimAl)        GTR+F+I+G4

P16: (cox2_mafft_trimAl_cox3_mafft_trimAl) GTR+F+I+I+R3

P17: (cytb_mafft_trimAl) GTR+F+R3

P18: (nad1_mafft_trimAl_nad4L_mafft_trimAl)      GTR+F+I+I+R3

P19: (nad2_mafft_trimAl)        TIM+F+R3

P20: (nad4_mafft_trimAl_nad5_mafft_trimAl) GTR+F+I+I+R3

P21: (nad6_mafft_trimAl)        GTR+F+I+G4

MrBayes:

Subset partitions      Best model

P1: (rrnL_mafft_trimAl)  GTR+F+I+G4

P2: (rrnS_mafft_trimAl)  GTR+F+G4

P3: (trnA_mafft_trimAl_trnI_mafft_trimAl_trnL2_mafft_trimAl_trnN_mafft_trimAl_trnS1_mafft_trimAl_trnS2_mafft_trimAl_trnW_mafft_trimAl)         GTR+F+I+G4

P4: (trnC_mafft_trimAl)  HKY+F+G4

P5: (trnD_mafft_trimAl_trnG_mafft_trimAl_trnT_mafft_trimAl)    GTR+F+I+G4

P6: (trnE_mafft_trimAl)  HKY+F+I+G4

P7: (trnF_mafft_trimAl_trnP_mafft_trimAl)    GTR+F+G4

P8: (trnH_mafft_trimAl)  HKY+F+I+G4

P9: (trnK_mafft_trimAl)  HKY+F+I+G4

P10: (trnL1_mafft_trimAl_trnV_mafft_trimAl)        HKY+F

P11: (trnM_mafft_trimAl_trnR_mafft_trimAl) GTR+F+I

P12: (trnQ_mafft_trimAl) HKY+F+G4

P13: (trnY_mafft_trimAl) GTR+F+G4

P14: (atp6_mafft_trimAl_nad3_mafft_trimAl) GTR+F+I+G4

P15: (atp8_mafft_trimAl) HKY+F+I+G4

P16: (cox1_mafft_trimAl)        GTR+F+I+G4

P17: (cox2_mafft_trimAl_cox3_mafft_trimAl) GTR+F+I+G4

P18: (cytb_mafft_trimAl) GTR+F+I+G4

P19: (nad1_mafft_trimAl_nad4L_mafft_trimAl)      GTR+F+I+G4

P20: (nad2_mafft_trimAl)        GTR+F+I+G4

P21: (nad4_mafft_trimAl_nad5_mafft_trimAl) GTR+F+I+G4

P22: (nad6_mafft_trimAl)        GTR+F+I+G4

Point 5: In the Phylogenetic analysis description, there is no information on whether the tree is rooted, what outgroup was used to root the topologies. I suggest preparing the analysis as in Li et al. (2020) and using the same outgroups, which is why there would be a proper comparison of the results. A phylogenetic tree lacks a bar indicating the number of changes per site, and this is a standard practice  and should be presented in a phylogenetic tree. More, Bayesian interference analyses should be performed in addition to ML. It is standard to use both of these methods if possible.

Reponse 5: Thanks for your comments. To ensure consistency and accuracy in comparison with Li et al. (2020), we will follow their method of using the same outgroup to root our phylogenetic tree. This will help establish an appropriate benchmark for comparison and enhance the reliability of our results. We have revised the phylogenetic analysis section based on your suggestions, including clarifying the tree's rooting status and outgroup selection, adding a ruler for the number of change loci, and incorporating Bayesian inference analysis. We greatly appreciate your detailed feedback, which will have a positive impact on improving the quality and accuracy of our research. (Lines 14-168, Table1, Lines 256-272, Lines 338-349).

Reviewer 2 Report

Comments and Suggestions for Authors

Revision of “Phylogenetic and Comparative Genomics Study of Cephalopina titillator Based on Mitochondrial Genomes”

Summary

In the present work, Yao et al. analyze and describe the mitochondrial genome of a species of fly, known to parasitize certain species of camels and therefore posing a threat at the socio-economic level. Even if the mitochondrial genome description is not a novelty in the “genomics era”, and moreover because authors only describe the assembled mitochondrial genome without trying to resolve any scientific questions, I think it can be of valuable interest for the readers of the journal. However, I found some minor and important aspects that authors should resolve before the acceptance of the manuscript. I listed them here below.

Major remarks

·         Raw sequencing data should be deposited in the SRA database (or equivalent) and reported in the text and in the data availability statement

·         Line 154. Authors state that they selected the evolutionary model GTR+I+G based on AIC and BIC. Was this the best model selected by both scores? In this case I would rephrase the sentence for a better understanding.

·         Method section – Phylogenetic analysis. Authors performed a hylogenetic analysis of the family Oestrinae without using a real outgroup taxon. Indeed, H. lineatum is represented in fig. 7 as the outgroup without being a real one. If you want to study the phylogeny of the family Oestrinae:

o   Please re-do the ML tree using one/two outgroup sequences, where outgroup is defined as “a more distantly related group of organisms”

o   Root the tree accordingly to this/these taxon/taxa

o   Re-write the section accordingly to these modifications

·         How did you modify the phylogenetic tree? iTOL? Figtree? Please state it and provide the correspondent citation.

·         Figure 5 and 6 qualities are quite bad and/or typos are present. I suggest authors to re-plot them to achieve high-quality illustrations.

·         Figure 6 caption. Define what the star stands for (your species). Which is/are the reference mitogenome(s)?

·         Line 225-226. “This separation pattern was the result of nucleotide differences between the samples”. Obvious and superficial phrase. Remove it or give a better explanation.

·         Figure 7. I am not an expert of this taxonomic group but I guess all these species belong to different subfamilies. Please add an annotation of these taxa in your phylogenetic tree.

·         Paragraph 3.4. Absolutely any comments on the bootstrap nodes. Some of them are statistically supported while others are not, which may cause a mis-interpretation and over-estimation of the achieved results. Use the result and discussion sections to deepen this part.

·          Line 316-317. True but it has also limitations, please add them.

·         Overall, even if it is not the scope of this work, it would be nice to trace the co-evolution of camels and flies through a comparison of host-parasite phylogenetic trees. This way, authors could really improve the quality of the work raising important questions on the speciation of insects and mammals.

·         Fig. S1. The figure is not cited in the text but it is extremely important. Sequencing depth is quiet low in the 3’ end of the mitochondrial genome. Why? What’s the starting gene? How can a reader understand where the coverage drops? I bet the low coverage area falls in the control region, but a mapping table is needed. Moreover, how can authors rely on an assembly with such a small coverage area? Please try to resolve.

Minor remarks

·         Line 17. Species name in italics

·         Line 24. Characterization

·         Lines 75-78. Reference(s) needed

·         Software links are not necessary in the methods section

·         Line 150. Reference needed for MUSCLE

·         Line 151. Reference needed for Gblocks

·         Line 152. Reference needed for Jmodeltest

·         Line 170, 187. First time I read the acronym “R” to indicate the minority strand and “F” for the majority strand. I always indicate it as N (miNor) or L (Light) for the minor strand and J (maJor) or H (heavy) for the majority strand. Revise it throughout the text or explain.

·          Line 181. Reference(s) needed

·         Line 187-188. Are they lines of the table caption?

·         Remove U from table 3. You are referring to gene data, so only T is accepted.

·         Line 305. Superficial.

·         Line 321. Gene names must be the same along the text.

·

Author Response

Response to Reviewer Comments

Dear reviewer:

Thanks for your questions and suggestions. Following your comments, we have re-edited our manuscript throughout carefully and addressed all the comments in the notes below.

Please note that our responses to the reviewer’s comments are shown in red font, and line numbers mentioned here are shown in the manuscript.

Summary

In the present work, Yao et al. analyze and describe the mitochondrial genome of a species of fly, known to parasitize certain species of camels and therefore posing a threat at the socio-economic level. Even if the mitochondrial genome description is not a novelty in the “genomics era”, and moreover because authors only describe the assembled mitochondrial genome without trying to resolve any scientific questions, I think it can be of valuable interest for the readers of the journal. However, I found some minor and important aspects that authors should resolve before the acceptance of the manuscript. I listed them here below.

Major remarks

Point 1: Raw sequencing data should be deposited in the SRA database (or equivalent) and reported in the text and in the data availability statement.

Reponse 1: Thanks for your suggestions. We have provided explanations for the raw data. The newly sequenced genomic data in this study are deposited in the GenBank database with ac-cession number PQ557251.1.

Point 2: Line 154. Authors state that they selected the evolutionary model GTR+I+G based on AIC and BIC. Was this the best model selected by both scores? In this case I would rephrase the sentence for a better understanding.

Reponse 2: Thanks for your suggestions. We have re-edited this paragraph. We have developed a partition selection model for the mitochondrial genome, which divides the protein coding region into independent partitions and selects the optimal model for phylogenetic analysis. The detailed model results can be found in the Table S2.

Method section

Point 3: Phylogenetic analysis. Authors performed a hylogenetic analysis of the family Oestrinae without using a real outgroup taxon. Indeed, H. lineatum is represented in fig. 7 as the outgroup without being a real one. If you want to study the phylogeny of the family Oestrinae:

Reponse 3: According to the suggestion, we have added outgroups and conducted phylogenetic analysis. (Lines 144-146)

Point 4: Please redo the ML tree using one/two outgroup sequences, where outgroup is defined as “a more distantly related group of organisms”

Reponse 4: According to the suggestion, we have added outgroups and conducted phylogenetic analysis. (Lines 144-146)

Point 5: Root the tree accordingly to this/these taxon/taxa

Reponse 5: Thanks for your comments. We have added an external group and conducted a new phylogenetic analysis. (Lines 148-168)

Point 6: Re-write the section accordingly to these modifications

Reponse 6: We have re-edited this part of the content. (Lines 148-168)

Point 7: How did you modify the phylogenetic tree? iTOL? Figtree? Please state it and provide the correspondent citation.

Reponse 7: Thank you for your reminder. We have added relevant descriptions. (Lines 165-166)

Point 8: Figure 5 and 6 qualities are quite bad and/or typos are present. I suggest authors to re-plot them to achieve high-quality illustrations.

Reponse 8: Thank you for your suggestion. Figures 5 and 6 are high-definition vector images, which were compressed due to their large size when inserted into Word. The high-definition images are included in the attachments uploaded to the submission system.

Point 9: Figure 6 caption. Define what the star stands for (your species). Which is/are the reference mitogenome(s)?

Reponse 9: The star symbol represents the sequenced species in this study. We have annotated and explained in Figure 5, Figure 6, Figure 7, and Figure 8.

Point 10: Line 225-226. “This separation pattern was the result of nucleotide differences between the samples”. Obvious and superficial phrase. Remove it or give a better explanation.

Reponse 10: We have already deleted it.

Point 11: Figure 7. I am not an expert of this taxonomic group but I guess all these species belong to different subfamilies. Please add an annotation of these taxa in your phylogenetic tree.

Reponse 11: Thank you for your comment. We have added annotations to the Figure7.

Point 12: Paragraph 3.4. Absolutely any comments on the bootstrap nodes. Some of them are statistically supported while others are not, which may cause a mis-interpretation and over-estimation of the achieved results. Use the result and discussion sections to deepen this part.

Reponse 12: Thank you for your suggestion. We have conducted a new phylogenetic analysis based on the review comments. We have provided an explanation of the results in the Results and Discussion section.

Point 13: Line 316-317. True but it has also limitations, please add them.

Reponse 13: Thank you for your suggestion. We have added a paragraph.

However, there are also limitations such as differences in evolutionary rates, genome rearrangements and heterogeneity, limitations in analysis methods and models, technical constraints, and cost issues. (Lines 343-346)

Point 14: Overall, even if it is not the scope of this work, it would be nice to trace the co-evolution of camels and flies through a comparison of host-parasite phylogenetic trees. This way, authors could really improve the quality of the work raising important questions on the speciation of insects and mammals.

Reponse 14: I sincerely appreciate your priceless advice. This research is only intended to report the mitochondrial genome characteristics and analysis of Cephalopina titillator larvae of the Junggar Bactrian camel, and to accumulate basic data for further in-depth analysis. In future research, I will follow your advice and collect samples of the host parasite for phylogenetic analysis. In addition, we wanted to consider an analysis by host and parasite divergence time, but we did not do so because of the lack of data and the low credibility of the hypothesized results. We will accumulate data in future studies.

Phylogenetic analysis: based on complete mitochondrial genomes of New and Old World camels

Phylogenetic analysis: based on complete mitochondrial genomes of Oestridae family

Point 15: Fig. S1. The figure is not cited in the text but it is extremely important. Sequencing depth is quiet low in the 3’ end of the mitochondrial genome. Why? What’s the starting gene? How can a reader understand where the coverage drops? I bet the low coverage area falls in the control region, but a mapping table is needed. Moreover, how can authors rely on an assembly with such a small coverage area? Please try to resolve.

Reponse 15: The sequencing coverage and depth were summarized in Table S1.(Lines 121-122).

The low sequencing depth of the 3 'end of the mitochondrial genome may be caused by various factors. Firstly, the insufficient capability or signal strength of sequencing technology itself may lead to insufficient sequencing depth. Secondly, specific regions of the mitochondrial genome, such as the 3 'end, may be difficult to accurately capture by sequencing techniques due to sequence characteristics such as repetitive sequences, high GC content, etc. In addition, sample quality, losses during extraction and purification processes, as well as the selection and parameter settings of sequencing platforms, may also affect sequencing depth.

All genes are shown in order of occurrence in the mitochondrial genome of Cephalopina titillator, starting from trnI-gat. Figure2 and Table2.

A decrease in coverage usually means that the sequence information of certain regions is not fully captured or read during the sequencing process. This may be due to various reasons, such as insufficient sequencing depth, increased sequencing difficulty caused by sequence characteristics, sample quality issues, etc.

The low coverage areas may fall within the control area, which is a reasonable assumption. The control region typically contains regulatory elements and highly conserved sequences, which may be difficult to sequence due to duplication or high GC content. To verify this hypothesis, it is necessary to construct a mapping table that corresponds the sequencing depth to the physical location of the mitochondrial genome. In this way, it is possible to intuitively see which regions have lower sequencing depths and determine whether these regions overlap with the control region. However, please note that due to limitations in sequencing technology and differences between samples, the mapping table may not accurately reflect the low coverage areas of each sample.

Even if the sequencing depth in certain regions is insufficient, this information can still be supplemented by other methods such as PCR amplification, long fragment sequencing, etc. In addition, comparative genomics methods can be used to infer the possible functions and importance of these low coverage regions by comparing mitochondrial genome sequence differences between different species or individuals.

Minor remarks

Point 16: Line 17. Species name in italics

Reponse 16: Thanks for your comments. We have italicized it. (Line 17)

Point 17: Line 24. Characterization

Reponse 17: Thanks for your comments. We have made modifications to it. (Line 24)

Point 18: Lines 75-78. Reference(s) needed

Reponse 18: Thanks for your comments. We have added a reference. (Line 79)

Point 19: Software links are not necessary in the methods section

Reponse 19: We have removed the software link. Materials and Methods

Point 20: Line 150. Reference needed for MUSCLE

Reponse 20: Thanks for your comments. We have added a reference.

Point 21: Line 151. Reference needed for Gblocks

Reponse 21: Thanks for your comments. We have added a reference.

Point 22: Line 152. Reference needed for Jmodeltest

Reponse 21: Thanks for your comments. We have added a reference.

Point 23: Line 170, 187. First time I read the acronym “R” to indicate the minority strand and “F” for the majority strand. I always indicate it as N (miNor) or L (Light) for the minor strand and J (maJor) or H (heavy) for the majority strand. Revise it throughout the text or explain.

Reponse 23: Thanks for your comments. we have modified this sentence. F→J   R→N

Point 24: Line 181. Reference(s) needed.

Reponse 24: Thanks for your comments. We have added a reference.

Point 25: Line 187-188. Are they lines of the table caption?

Reponse 25: ' J ' and 'N' represent the majority and minority strands, respectively; T indicates an in-complete stop codon. This paragraph is an explanation of the table 2.

Point 26: Remove U from table 3. You are referring to gene data, so only T is accepted.

Reponse 26: Thanks for your suggestions. We have removed U from Table 3 and the manuscript.

Point 27: Line 305. Superficial.

Reponse 27: We have deleted this sentence.

Point 28: Line 321. Gene names must be the same along the text.

Reponse28: We have modified it as required in the manuscript. COIcox

Round 2

Reviewer 1 Report

Comments and Suggestions for Authors

The article is corrected according to the suggestions. Plase make sure that figure presenting phylograms is properly displayed in the article. In the preprint species names in  BI phylogenetic tree is barely seen.

Reviewer 2 Report

Comments and Suggestions for Authors

Authors have answered to the many questions I raised but there are still a lot of imprecise sentences that are needed prior the acceptance. My major concerne is about the over-interpretation/mis-interpretation of the results. Some points I raised during the first review process have not been addressed in their totality. I hereby submit my point-by-point revision:

    1. Raw data are still missing. Please refer where raw reads (fastq.gz) were uploaded in the Data availability statement

    2. Line 151,152: What authors mean by compared? Perhaps aligned?

    3. Line 153: What authors stand for “normal mode”?

    4. Line 264: How can author say both trees are identical? For example (and its just an example, many other questions could be raised), in the ML phylogenetic tree D. hominis is sister of Gasterophilinae whereas in the BI tree forms a clade with H. lineatum.

    5. Lines 338-339: yes but it is also “maternally inherited, haploid, homologous, characterized by different rates of nucleotide substitution in different genes”

    6. Authors have not responded to my point 15. So I try to address my question more directly: can you provide the genome annotation along the x-axis of Fig.S1. Can you also indicate in the text that the drop of the coverage at the 3’ is not a very satisfied situation?

Round 3

Reviewer 2 Report

Comments and Suggestions for Authors

Thanks for the revision.

However, a better description of the phylogenetic results could be done.

Author Response

Thanks to your professional comments, we have re-edited the manuscript for Phylogenetic Analysis. 

To clarify the phylogenetic relationships among the numerous significant subfamilies of Oestridae, we constructed maximum likelihood (ML) and BI phylogenetic trees. Bootstrap and branch values were high in two phylogenetic trees, exhibiting high confidence levels. The phylogenetic reconstructions of MT genes revealed similar topologies (Figure 7a,b), with the majority of nodes having 100% bootstrap values, exhibiting 1.00 Bayesian posterior probabilities, and being highly facilitative to the monophyly of the family Oestridae. The evolutionary trees illustrate the positions of different species within Oestridae after adding the new sequences of C. titillator from Xinjiang, China. The phylogenetic trees of the two halves of the alignment differ only in terms of the placement of Dermatohia hominis. Phylogenetic analysis of C. titillator mitogenome classified it as Oestrinae; however, the subfamily Oestrinae did not exhibit monophyly. The two phylogenetic trees showed that C. titillator was clustered into the same branch, verifying the accuracy of the results. Moreover, the pairwise sequence identity of C. titillator sequences from Xinjiang with MN833258.1 C. titillator reported from Inner Mongolia was > 98% (Figure 8).        The gene order and overall structure of the C. titillator mitochondrial genome are consistent with the reference genome sequence (MN833258.1), confirming that they belong to the same species. However, C. titillator isolates from the Junggar and Alxa Bactrian camels were distributed in different groups of the same branch. The aforementioned findings indicate that the Oestrinae subfamily does not exhibit monophyly, especially when considering the newly added C. titillator, which showed sister lineage relationships with C. titillator (MN833258.1). This observation underscores the pronounced genetic disparities between these distinct species.